# Association of the APOA-5 Genetic Variant rs662799 with Metabolic Changes after an Intervention for 9 Months with a Low-Calorie Diet with a Mediterranean Profile

**DOI:** 10.3390/nu14122427

**Published:** 2022-06-11

**Authors:** Daniel de Luis Roman, David Primo, Olatz Izaola, Rocio Aller

**Affiliations:** Center of Investigation of Endocrinology and Nutrition, Department of Endocrinology and Investigation, Medicine School, Hospital Clinico Universitario, University of Valladolid, 47003 Valladolid, Spain; dprimo@yahoo.es (D.P.); oizaola@yahoo.es (O.I.); raller@yahoo.es (R.A.)

**Keywords:** *ApoA5* gene, rs662799, low-calorie diet, triglycerides

## Abstract

In cross-sectional studies, the genetic variant rs662799 of the APOA5 gene is associated with high serum triglyceride concentrations, and in some studies, the effect of short-term dietary interventions has been evaluated. The aim of the present investigation was to evaluate the role of this genetic variant in metabolic changes after the consumption of a low-calorie diet with a Mediterranean pattern for 9 months. A population of 269 Caucasian obese patients was recruited. Adiposity and biochemical parameters were measured at the beginning (basal level) and after 3 and 9 months of the dietary intervention. The rs662799 genotype was assessed with a dominant analysis (TT vs. CT + CC). The APOA5 variant distribution was: 88.1% (*n* = 237) (TT), 11.5% (*n* = 31) (TC) and 0.4% (*n* = 1) (CC). There were significant differences only in triglyceride levels at all times of the study between the genotype groups. After 3 and 9 months of dietary intervention, the following parameters improved in both genotype groups: adiposity parameters, systolic pressure, total cholesterol, LDL cholesterol, leptin, adiponectin and the leptin/adiponectin ratio. The intervention significantly decreased insulin levels, HOMA-IR and triglyceride levels in non-C allele carriers (Delta 9 months TT vs. TC + CC). i.e., insulin levels (delta: −3.8 + 0.3 UI/L vs. −1.2 + 0.2 UI/L; *p* = 0.02), HOMA-IR levels (delta: −1.2 + 0.2 units vs. −0.3 + 0.1 units; *p* = 0.02), triglyceride levels (delta: −19.3 + 4.2 mg/dL vs. −4.2 + 3.0 mg/dL; *p* = 0.02). In conclusion, non-C allele carriers of rs662799 of the APOA5 gene showed a decrease of triglyceride, insulin and HOMA-IR levels after consuming a low-calorie diet with a Mediterranean pattern; we did not observe this effect in C allele carriers, despite a significant weight loss.

## 1. Introduction:

Dyslipidemia is a metabolic problem commonly detected in obese subjects [1], and obese individuals usually presents elevated triglyceride levels and decreased HDL-cholesterol, secondarily associated with atherosclerosis [2]. The apolipoprotein A5 (*ApoA5*) gene is located on chromosome 11q23.3 in the *APOA1/C3/A4A5* gene cluster. This gene encloses four exons coding for a 366-amino acid protein called *APOA5*. This apolipoprotein is secreted from the liver, regulates triglyceride metabolism and is part of HDL–cholesterol particles [3,4]. The role of *APOA5* in the metabolism of triglycerides involves different pathways; this protein reduces the synthesis of very low density lipoprotein (*VLDL*) triglycerides and increases the hydrolysis of *VLDL* triglycerides, secondary to the action of lipoprotein lipase [3,4]. Some studies have reported that the *APOA5* gene is clearly associated with the establishment of cardiovascular events [5] and metabolic syndrome [6,7].

Moreover, single-nucleotide variants (SNVs) in the *APOA5* gene, such as rs651821 and rs3135506, have been associated with metabolic syndrome and its traits, [8,9,10,11,12]; recently, the involvement of rs662799 (−1131T > C) [13] in lipid profile and metabolic syndrome has been shown too. This last genetic variant of the APOA5 gene has been described to be a functional-tag SNV [11,12]. The C allele of rs662799 deteriorates ribosomal translation efficiency, and this decreases the concentrations of APOA5 and decreases lipoprotein lipase activity [8,9]. In the literature, there are few intervention studies evaluating the role of this SNV, but they have shown some interesting findings. In a cross-sectional design with hypertriglyceridemic patients, C allele carriers reported higher levels of triglycerides than non-C allele carriers, without a relation to total energy intake and total saturated and polyunsaturated fat intake [14]. After a well-designed normocaloric dietary changes during 4 weeks in hypertriglyceridemic subjects, only carriers of the T allele showed an improvement in the lipid profile [15]. Another short-term study (3 months) with a high-carbohydrate/low-fat hypocaloric diet (1500 cal/day) reported that postprandial triglyceride elevation is modulated by the T allele [16]. In other short-term (3 months) dietary intervention with a hypocaloric diet (1500 cal/day) with a Mediterranean profile, C allele carriers showed a lack of response after weight loss of the triglyceride levels [17]. Finally, another cross-sectional design [18] reported a relationship between the C allele and a high fat intake; it reported a relationship between triglyceride levels and a high dietary fat intake in subjects with the minor allele.

At this moment, there are no long-term nutritional interventions that have evaluated changes in the relationship of rs662799 with the metabolic. This is an interesting investigation topic, since it was shown that weight loss following a Mediterranean dietary pattern decreased the lipid levels [14]. The Mediterranean diet is a primarily plant-based eating plan that includes the daily intake of whole grains, olive oil, fruits, vegetables and legumes, nuts, herbs, and spices. Other foods like animal proteins are eaten in small quantities, with the animal protein being preferentially obtained from fish and seafood. These types of foods contain high amounts of dietary unsaturated fats and have potential effects on the lipid profile s they may affect directly the lipids levels [15,16].

The aim of our study was to evaluate the role of SNV rs662799 on metabolic variables in obese subjects after a low-calorie diet with a Mediterranean pattern during a period 9 months.

## 2. Subjects and Methods

### 2.1. Subjects and Clinical Investigation

The present study enrolled a total of 269 unrelated obese subjects from our health area during a routine check-up of obesity, with a body mass index >30 kg/m^2^. The selection method was consecutive non-probabilistic. All subjects agreed to enroll in the investigation and provided written informed consent. Inclusion criteria were body mass index >30 kg/m^2^ and age > 18 years. Exclusion criteria were the presence of any associated condition, coronary events, renal or hepatic disorders, alcohol intake more than 20 g/day in women and more than 30 g/day in men or within the 12 months before the study, and receiving medications known to influence the lipid levels (statins, fibrates, hormonal therapy, glucocorticoids and anti-inflammatory drugs).

All measurements were realized in our department of investigation by physicians. Anthropometric parameters were measured for all subjects for body weight, height, body mass index (BMI) and waist circumference at the beginning and at 3 and 9 months of dietary intervention. Total body fat mass was measured by electric impedance determination. For the analysis of biochemical parameters and the genotype of rs662799, 15 mL of venous blood was drawn and aliquoted in ethylenediaminetetraacetic acid (EDTA)-coated probes after 10 h of overnight fast. Lipid profile, C reactive protein (CRP), adipokine levels (leptin, total adiponectin, resistin and the adiponectin/leptin ratio), fasting glucose, insulin were measured, and insulin resistance was calculated with the homeostasis model assessment (HOMA-IR) method.

### 2.2. Adiposity Parameters and Blood Pressure

Height (cm) was determined using a non-elastic measuring tape (Quimild, Spain). Waist circumference (cm) was measured using a non-elastic measuring tape (Quimild, Spain). Body weight was measured using a digital scale (Quimild, Spain) and recorded to the nearest 10 g. Body mass index (BMI) was determined with the equation (body weight (in kg) divided by height (in m^2^)). Total fat mass was measured by impedance with an accuracy of 5 g [19] (EFG BIA 101 Anniversary, Akern, It), and this parameter was calculated with the equation (0.756 Height2/Resistance) + (0.110 Body mass) + (0.107 Reactance) − 5.463.

Blood pressures (systolic and diastolic) were determined by averaging four measurements (Omrom, Los Angeles, CA, USA), after a rest time of 10 min.

### 2.3. Analytical Parameters

Glucose concentrations were determined by an automated hexoquinase oxidase system (COBAS INTEGRA 400 analyzer (Roche Diagnostic, Montreal, QC, Canada). Insulin concentrations was determined by an electrochemiluminescence assay (COBAS INTEGRA 400 analyzer (Roche Diagnostic, Montreal, QC, Canada). Insulin resistance was calculated with the formula of the homeostasis model assessment for insulin resistance (HOMA-IR) (glucose x insulin/22.5) [20]. Different components of the lipid profile (triglycerides, total cholesterol and HDL–cholesterol) were measured using the COBAS INTEGRA 400 analyzer (Roche Diagnostic, Montreal, QC, Canada). LDL cholesterol was obtained in an indirect way through the Friedewald formula (LDL cholesterol = total cholesterol-HDL cholesterol-triglycerides/5) [21]. C-reactive protein (CRP) was measured by immunoturbimetry (Roche Diagnostics GmbH, Mannheim, Germany).

All adipokines were determined by the Enzyme-Linked Immuno Sorbent Assay (ELISA) as follows: leptin (Diagnostic Systems Laboratories, Inc., Webster, TX, USA) (DSL1023100) [22], adiponectin (R&D systems, Inc., McKinley, MN, USA) (DRP300) [23] and resistin (Biovendor Laboratory, Inc., Brno, Czech Republic) (RD191016100) [24].

### 2.4. Genotyping of the APOA5 Gene Variant

DNA was isolated from leucocytes, using a commercially available DNA isolation kit (Biorad, Los Angeles, CA, USA). Genotyping was realized by polymerase chain reaction real-time analysis. Primers were designed with the Sequenom Assay Design v4 (SEQUENOM, Inc. San Diego, CA, USA). The polymerase chain reaction (PCR) was realized with 50 ng of genomic DNA and 0.25–0.30 µL of each oligonucleotide primer for rs662799 (primer forward: 5′-GAGCCCCAGGAACTGGAGCGAAAGT–3′ and reverse 5′–AGATTTGCCCCATGAGGAAAAGCTG–3′, in a 3.0 µL final volume (Lifetecnologies, LA, CA). DNA was denatured at 90 °C for 10 min; subsequently, 45 cycles at 65 °C for 15 s, and annealing at 59 °C for 40 s were performed. PCRs were run in a 2.5 µL volume containing 0.15 µL of iPLEx Termination mix (Bio-Rad^®^, San Diego, CA, USA) with hot start Taq DNA polymerase. Then, 15% of the samples were randomly re-evaluated to confirm accuracy.

### 2.5. Dietary Intervention

Obese subjects were assigned to consume a low-calorie diet (around 1000 calories per day) with a pattern of a Mediterranean diet for 9 months. The target distribution of macronutrients in this diet was 53% carbohydrates, 27% fats, and 20% proteins. The target percentage of dietary fats was 65% of monounsaturated fats, 20% of saturated fats, and 15% of polyunsaturated fats. Food tables were utilized, including legumes, vegetables, poultry, whole grains, fish, fresh fruit, olive oil and limiting unhealthy fats such as margarines, fatty meats, snacks, industrial pastries. All the subjects were given written instructions by a registered dietitian on completion of a 3-day (1-weekend and 2-week days) dietary record and basal time and after 3 and 9 months of dietary intervention. These records were analyzed with a software (Dietsource^®^, Geneve, Swittzerland) with food references for our country [25]. All participants attended four sessions (30 min each, with example menu plans) at the start of the trail and during the intervention to explain the diet. The physical exercise activity indicated included sessions of aerobic exercise at least 3 times per week (45–60 min each), and the patient recorded it on a self-reported questionnaire. All subjects were interviewed weekly by phone to evaluate whether they were following the Mediterranean diet and the physical exercise activity plan.

### 2.6. Statistical Analysis

The sample size (*n* = 250) was obtained to determine differences over 10 mg/dL of triglyceride concentrations with 90% power and 5% significance. Statistical evaluation was performed using SPSS for Windows, version 23.0 software package (SPSS Inc. Chicago, IL, USA). We performed the analysis with a dominant genetic model (TT vs. TC + CC). Descriptive statistics are presented as mean ± standard deviation for continuous variables and as a percentage for categorical variables. Two-tailed Student’s *t*-test was used to analyze continuous variables with normal distribution. Chi-square test, with Yates correction as necessary, was used to analyze categorical variables. In order to reduce Type I errors in the association analysis, the Bonferroni test was applied for multiple testing. The statistical analysis to evaluate the interaction between the gene and the dietary intervention was performed using ANCOVA (covariance analysis) adjusted by age, sex, and BMI modeling the dependent variable with the starting values. Hardy–Weinberg equilibrium was determined with Chi-square test to compare our expected and observed data.

### 2.7. Ethical Approval

All procedures were in accordance with the ethical standards of the institutional research committee (HVUVA committee 7/2020) and with the 1964 Helsinki declaration and its later amendments. Written informed consent was obtained from all individual participants included in the study.

## 3. Results

The rs662799 distribution was as follows: 88.1% (*n* = 237) of the participants (TT) were homozygous for the T allele, 11.5% (*n* = 31) (TC) were heterozygous, and 0.4% (*n* = 1) (CC) were homozygous for the C allele. Finally, the T allele frequency was 0.94 and the C allele frequency was 0.06; therefore, throughout the analysis we used a dominant evaluation (TT vs. TC + CC). This variant was in Hardy–Weinberg equilibrium (*p* = 0.43).

The average age of our sample was 51.3 + 5.2 years (range: 25–66,), and the average BMI was 40.7 + 3.2 kg/m^2^ (range:33.3–43.4). Gender distribution was 199 females (74.0%) and 70 males (26.0%). The average age of both genotype groups was similar (TT; 51.9 + 4.9 years vs. TC + CC; 51.0 + 4.2 years: ns) as well as the gender proportion (TT 25.7% males vs. 74.3% females vs. TC + CC 28.1% males vs. 71.9% females). Patients followed the caloric restriction and macronutrient recommendations as indicated by the dietitian (Table 1). The percentages of macronutrients were those indicated in the initial objectives, and the same happened with the total fat intake and the amount of unsaturated and saturated fats. Basal and post-intervention physical exercise levels were similar in both groups (Table 1).

### 3.1. Modifications in Anthropometry and Blood Pressure

Adiposity variables and systolic/diastolic pressure were not significantly related to the rs662799 variant, as shown in Table 2. When we compared both genotype groups at baseline, there were no significant differences in any parameter. After the dietary intervention during a period of 9 months and in both genotype groups, all the following parameters improved at 3 and 9 months (Delta 9 months TT vs. TC + CC); BMI (delta:−4.5 + 0.5 kg/m^2^ vs. −4.4 + 0.3 kg/m^2^; *p* = 0.51), body weight (delta:−6.6 + 2.0 kg vs. −6.4 + 1.9 kg; *p* = 0.43), fat mass (delta:−5.1 + 0.8 kg vs. −5.0 + 1.1 kg; *p* = 0.59), waist circumference (delta:−6.2 + 1.9 cm vs. −6.0 + 1.8 cm; *p* = 0.51) and systolic pressure (delta:−4.8 + 1.0 mmHg vs. −5.2 + 2.0 mmHg; *p* = 0.49). These changes were the same in the two genotype groups.

### 3.2. Changes in Classical Biochemical Parameters

When we compared the biochemical parameters in the two groups at baseline, only the triglyceride (TG) levels showed significant differences. These TG concentrations were higher in patients with the C allele (TT vs. TC + CC in basal levels: −19.9 + 3.1 mg/dL; *p* = 0.02), (TT vs. TC + CC in post treatment 3 months levels: −20.1 + 3.2 mg/dL; *p* = 0.03) and (TT vs. TC + CC in post treatment 9 months levels: −34.7 + 5.8 mg/dL; *p* = 0.03), see Table 3.

Considering glucose metabolism after the dietary intervention (Delta 9 months TT vs. TC + CC), insulin concentrations (Delta: −3.8 + 0.3 UI/L vs. −1.2 + 0.2 UI/L; *p* = 0.02) and HOMA-IR (Delta: −1.2 + 0.2 units vs. −0.3 + 0.1 units; *p* = 0.02) decreased significantly in non-C allele carriers. After the dietary intervention (Delta 9 months TT vs. TC + CC), TG concentrations (Delta: −19.3 + 4.2 mg/dL vs. −4.2 + 3.0 mg/dL; *p* = 0.02) improved in non-C allele carriers, too. Total cholesterol and LDL cholesterol concentrations decreased in both genotype groups.

### 3.3. Adipokine Levels

Table 4 reports changes of serum adipokines and the adiponectin/leptin ratio. The serum levels of these adipokines were similar in TT and TC + CC genotype groups, in baseline values and in levels throughout the intervention. After the dietary intervention and in both genotypes (delta 9 months TT vs. TC + CC), the serum adiponectin levels increased (delta: 24.2 + 3.1 ng/dL vs. 21.9 + 3.9 ng/dL; *p* = 0.42), and the leptin levels decreased (delta: −19.2 + 5.1 ng/dL vs. −20.1 + 4.1 ng/dL; *p* = 0.39). Finally, the calculated adiponectin/leptin ratio increased in both genotypes (delta 9 months TT vs. TC + CC:0.55 + 0.2 vs. 0.61 + 0.1 ng/dl; *p* = 0.49). The resistin levels remained unchanged.

## 4. Discussion

The results of this interventional study in Caucasian obese subjects showed that the single-nucleotide variant (SNV) rs662799 of the APOA5 gene modulated the metabolic response of weight change secondary to a long-term low-calorie diet with a Mediterranean pattern. In C allele carriers, this dietary intervention generated significantly lower improvements in insulin, HOMA-IR and triglyceride levels than in non-C allele carriers.

Investigations in the literature evaluating the role of this SNV on metabolic modifications are scarce and with a very short duration. For example, in the literature there is only one study that evaluates the role of rs662799 in the response to statins [26]. This study reported that dislipidemic subjects with the C allele benefited less from statin treatment compared with non-C allele carriers. Despite these few studies, this SNV has been associated with high levels of triglycerides (TG) [27] and a high occurrence of cardiovascular events [28,29]. These scientific findings can be explained by the fact that the C allele reduces ribosomal translation activity, thereby decreasing the concentrations of APOA5 [30], and this fact negatively affects lipoprotein lipase activity and raises serum TG levels [31].

The relationship of different diets with this SNV have been evaluated, although in short-term interventions. In a previous study with borderline to moderate hypertriglyceridemic Chinese patients [15], Jang et al. showed that a dietary intervention during 3 months decreased the triglyceride levels in TT subjects, with a lack of action in C allele carriers. The dietary changes consisted in increasing the vegetable amount to at least 180–420 g/day and substituting by 1/3 the intake of refined rice with legumes three times per day, as a carbohydrate source. The decrease in TG levels in this study was 20% secondary to changes in serum APOA5 concentration. In our study the improvement was 16%, very similar to that of the study previously discussed [15]. In this previous work, insulin resistance was not evaluated, and also there were differences in the design with respect to our work; in fact, a low-calorie diet was not used, the nutritional intervention was shorter, while the evaluated subjects were not normolipemic obese as in our case. In another short-term intervention trial of 6 days [16], healthy Korean volunteers followed a high-carbohydrate/low-fat diet, in which the percentages of carbohydrates, proteins and fats were 70%, 15% and 15%, respectively. The energy intake of the patients was dependent on their satiety. Subjects with the C allele had higher TG levels after this short intervention trial than TT allele subjects. An intervention trial of 3 months with a hypocaloric Mediterranean diet in a Caucasian population [17] reported a decrease in HOMA-IR, insulin levels and triglycerides after a significant weight loss in non-C allele carriers, with a lack of metabolic effect in C allele carriers. In this study, there was a caloric restriction of 500 calories per day, and it was carried out in obese subjects. The dietary distribution was 50% of the calorie value from carbohydrates, 30% from lipids and 20% from proteins. The proportion of fats was 55% from monounsaturated fats, 30% from saturated fats and 15% from polyunsaturated fats. The caloric restriction in our present study was higher than that in this previous study [17], while the distribution of macronutrients was similar, with a Mediterranean pattern, though with a higher percentage of monounsaturated fats. Despite these differences in the study design, the metabolic results of both studies are similar, with significant improvements in triglyceride, HOMA-IR and insulin levels in the non-carriers of the C allele. The postprandial state can play a relevant role in the above-mentioned findings, as Kim et al. [18] reported using a meal tolerance test. In non-obese healthy Korean subjects, C allele carriers showed a deferred peak time of TG and higher postprandial chylomicron levels and free fatty acids (FFA) and insulin levels after a high-fat meal compared to a low-fat meal [18]. This decreased clearance of TG in carriers of the C allele may produce a decrease in APOA5 function. In the postprandial situation, the delivery of FFA into the circulation is modulated by the intracellular hormone-sensitive lipase (HSL) and the way that lipoprotein lipase-derived fatty acids are not captured by adipose tissue and muscle [32]. Therefore, this metabolic change detected in the postprandial state after a high-fat meal could be secondary to a decrease of the postprandial suppression of HSL by insulin in C allele carriers, due to an insulin resistance state. This fact could be related to the lack of response to insulin and HOMA-IR in patients carrying the C allele in our study after the dietary intervention. In agreement of this hypothesis, recent investigations in Caucasian healthy subjects [33] reported that this SNV modulated postprandial lipidemia after a breakfast containing 75 g of fat, though the addition of glucose (25 g) to the tested breakfast suppressed this change. Perhaps, the secretion of insulin after a glucose overload could explain these findings, because insulin action was shown to decrease APOA5 expression and even to downregulate ApoA-5 levels in plasma [34].

In our present study in a Caucasian population, we used a diet with a Mediterranean profile based on its beneficial findings reported in other observational studies. Sanchez- Moreno et al. [35] detected a genotype–dietary fat interaction involving this genotype for obesity phenotypes in a Mediterranean population in a cross-sectional study. Hubacek et al. [36] reported that TG levels were higher in C allele carriers with the higher energy intake than in non-C allele carriers. Lai et al. [37] demonstrated that the n-6 (but not n-3) polyunsaturated fatty acids intake increased the fasting triglyceride concentrations in C allele carriers; therefore, the quality of the fat in the Mediterranean diet can also have an influence. Beyond the caloric restriction of the diet, some foods of this diet, such as lean meats, fish and olive oil with large amounts of polyphenols, monounsaturated fats, polyunsaturated, minerals and vitamins, could also explain the benefits observed in the metabolic parameters in non-C allele carriers [14]. Moreover, other nutrients could play an important role, for example, carriers of the minor allele with a low calcium intake (<500 mg/day) showed elevated triglyceride concentrations, in a cross-sectional study [38].

There are some limitations in our investigation. Firstly, the dietary intake was based on self-reports from the patients; these self-reported data could overestimate or underestimate the actual dietary intake. Second, considering the small allelic frequency of the C allele, the data of the genetic analyses should be evaluated with caution. Thirdly, the study was designed for Caucasian obese subjects aged 20–60 years without dyslipidemia and with a high BMI, so the data are not generalizable to the healthy population. Fourthly, our sample consisted of Caucasian obese patients, and our results could be generalized only to this subpopulation. Fifthly, the absence of a healthy control group decreases the power of the results obtained with our design. Sixthly, we used the same caloric restriction in males and females, and perhaps this is a potential bias, too. Finally, the circulating levels of ApoA5 were not been determined, and the determination of this apolipoprotein would have helped us to interpret our results also from a physiological perspective.

## 5. Conclusions

In conclusion, this long-term study demonstrated that the minor C allele of the APOA5 gene (rs662799) is negatively related to the effect on triglycerides, insulin levels and HOMA-IR after consumption of a low-calorie diet with a Mediterranean pattern. Further studies are necessary to evaluate the importance of this SNV in the lipid profile and metabolic syndrome in different populations [39]. In the future, clinical guidelines will incorporate this knowledge to improve the design of nutritional interventions for obese patients, using panels of SNVs and other omic technologies [40].

## Figures and Tables

**Table 1 nutrients-14-02427-t001:** Average daily intakes and physical activity determined at three times (basal and 3 and 9 months after the intervention) (mean ± SD).

TT (*n* = 237)				TC + CC (*n* = 32)		
	0 Time	3 Months	9 Months	0 Time	3 Months	9 Months
Calorie intake (kcal/day)	1728.9 + 210.8	1012.1 + 21.1 *	1008.1 + 13.1 *	1823.9 + 129.1	1010.1 + 18.1 *	1001.9 + 12.1 *
Carbohydrate intake (g/day) (PTC%)	207.8 + 51.9 (48.9%)	137.5 + 20.1 $ (54.3%)	130.1 + 18.1 $ (53.9%)	218.1 + 41.1 (47.8%)	133.1 + 19.1 $ (53.7%)	130.9 + 12.1 $ (52.6%)
Fat intake (g/day) (PTC%)	58.9 + 10.3 (30.6%)	31.1 + 8.1 # (26.7%)	32.2 + 5.1 # (26.9%)	62.4 + 8.3 (30.8%)	30.2 + 4.3 # (26.9%)	31.1 + 5.1 # (27.4%)
Protein intake (g/day) (PTC%)	88.6 + 14.1 (20.5%)	50.6 + 7.3 & (20.0%)	51.9 + 7.1 & (20.2%)	92.9 + 10.1 (22.4%)	51.2 + 6.9 & (20.5%)	52.3 + 6.0 & (21.0%)
Saturated Fat intake (g/day) (PFC%)	36.2 + 6.0 (61.4%)	6.1 + 4.9 ** (19,6%)	6.4 + 4.0 ** (19,0%)	37.8 + 5.1 (60.5%)	5.9 + 2.2 (19.5%)	5.3 + 1.1 (19.1%)
Monounsaturated Fat intake (g/day) (PFC%)	15.2 + 3.1 (25.8%)	21.3 + 2.9 $$ (63.2%)	22.1 + 2.0 $$ (64.1%)	15.8 + 5.2 (25.3%)	19.9 + 4.0 (65.9%)	20.1 + 4.1 (66.1%)
Polyunsaturated Fat intake (g/day) (PFC%)	7.5 + 6.0 (13.8%)	5.3 + 4.9 ## (17.2%)	5.4 + 3.2 ## (16.9%)	8.8 + 4.2 (14.2%)	4.4 + 2.2 (15.6%)	4.7 + 0.9 (15.8%)
Physical activity (min/week)	121.2 + 12.1	127.8 + 12.5	130.1 + 9.9	125.9 + 4.2	132.1 + 11.2	133.2 + 10.2

PTC: Percentage of total calories; PFC: Percentage of fat calories. Statistical differences *p* < 0.05, in each genotype group (* Daily Calorie intake, $ Daily Carbohydrate intake, # Daily fat intake, & Daily protein intake, ** Saturated fat intake, $$ monounsaturated fat intake, ## Polyunsaturated fat intake).

**Table 2 nutrients-14-02427-t002:** Adiposity parameters and pressure at three times (basal and 3 and 9 months after the intervention) (mean ± SD).

TT (*n* = 237)				TC + CC (*n* = 32)		
	0 Time	3 Months	9 Months	0 Time	3 Months	9 Months
BMI	40.8 ± 2.1	38.3 ± 3.2 *	36.1 ± 1.1 *	40,6 ± 2.3	38.4 ± 2.1 *	36.2 ± 1.0 *
Weight (kg)	91.4 ± 3.2	86.9 ± 2.0 $	84.8 ± 2.0 $	90.9 ± 2.1	86.8 ± 2.0 $	84.5 ± 2.1 $
Fat mass (kg)	36.3 ± 1.1	33.1 ± 1.0 #	31.2 ± 1.1 #	35.9± 1.2	33.0± 1.8 #	30.9 ± 1.8 #
WC (cm)	111.7 ± 4.3	107.1 ± 3.1 &	105.5 ± 3.0 &	110.8 ± 6.0	107.6 ± 4.1 &	104.9 ± 3.1 &
SBP (mmHg)	127.0 ± 3.0	123.1 ± 3.1 **	122.8 ± 4.0 **	126.9± 4.1	122.1 ± 3.1 **	121.7 ± 2.9 **
DBP (mmHg)	81.3 ± 3.1	78.8 ± 4.1	79.1 ± 4.0	81.4 ± 3.0	78.1 ± 3.0	78.9 ± 4.1

BMI: body mass index; DBP, diastolic blood pressure; SBP, systolic blood pressure; WC, waist circumference; Statistical differences *p* < 0.05, in each genotype group with basal value (* BMI, $ Weight, # fat mass, & WC, ** SBP). No statistical differences between genotype groups.

**Table 3 nutrients-14-02427-t003:** Biochemical parameters at three times (basal and 3 and 9 months after the intervention) (mean ± SD). (mean ± SD).

TT (*n* = 237)				TC + CC (*n* = 32)		
	0 Time	3 Months	9 Months	0 Time	3 Months	9 Months
Glucose (mg/dL)	103.8 ± 4.0	100.8 ± 3.1	99.8 ± 4.2	101.3 ± 6.0	99.9 ± 4.2	98.7 ± 4.1
Total ch. (mg/dL)	208.8 ± 4.6	195.2 ± 4.1 $	193.1 ± 3.1 $	208.7 ± 4.1	194.1 ± 3.1 $	192.1 ± 3.0 $
LDL-ch. (mg/dL)	137.2 ± 8.1	120.8 ± 2.2 #	118.1 ± 4.0 #	140.5 ± 3.9	122.5 ± 4.2 #	118.9 ± 3.1 #
HDL-ch. (mg/dL)	55.8 ± 1.6	54.7 ± 1.9	55.1 ± 1.2	55.9 ± 2.1	55.5 ± 1.3	56.1 ± 1.2
TG (mg/dL)	123.2 ± 11.0	110.9 ± 9.2 *	104.1 ± 6.2 *	143.1 ± 10.2 x	140.8 ± 4.1 x	139.1 ± 7.8 x
Insulin (mUI/L)	10.8 ± 2.1	9.2 ± 1.3 &	7.0 ± 1.0 &	12.3 ± 2.1	10.8 ± 2.2	11.0 ± 2.3
HOMA-IR	3.7 ± 1.0	2.1 ± 0.9 **	1.5 ± 0.7 **	3.9 ± 0.8	3.5 ± 1.1	3.6 ± 1.2
CRP (mg/dL)	4.3 ± 1.0	4.3 ± 1.3	4.2 ± 1.1	4.4 ± 2.1	4.5 ± 1.3	4.3 ± 1.1

HOMA-IR (homeostasis model assessment). CRP (C-reactive protein). Statistical differences *p* < 0.05, in each genotype group with basal values (total cholesterol $, LDL cholesterol #, triglycerides *, insulin &, HOMA IR **). Statistical differences *p* < 0.05, between genotype groups (triglycerides x).

**Table 4 nutrients-14-02427-t004:** Circulating adipocytokines at three times (basal and 3 and 9 months after the intervention) (mean ± SD). (mean ± S.D).

TT (*n* = 237)				TC + CC (*n* = 32)		
	0 Time	3 Months	9 Months	0 Time	3 Months	9 Months
Adiponectin (ng/mL)	4.8 ± 1.0	4.7 ± 2.0	4.9 ± 1.8	4.5 ± 2.0	4.4 ± 1.3	4.7 ± 1.2
Resistin (ng/mL)	39.1 ± 5.0	68.1 ± 4.2 $	63.1 ± 4.1 $	34.9 ± 4.1	67.2 ± 4.2 $	66.1 ± 3.1 $
Leptin(ng/mL)	88.1 ± 8.1	71.2 ± 4.1 *	68.9 ± 4.1 *	89.1 ± 8.1	70.1 ± 3.1 *	69.9 ± 6.1 *
Ratio Adiponectin/leptin	0.36 ± 0.2	0.92 ± 0.1 #	0.91 ± 0.1 #	0.18 ± 0.1	0.92 ± 0.3 #	0.93 ± 0.2 #

Statistical differences *p* < 0.05, in each genotype group (* leptin $ adiponectin, # ratio adiponectin/leptin). No statistical differences between genotype groups.

## Data Availability

The data that support the findings of this study are available from the corresponding author upon reasonable request.

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
