# Peer review of "Association of the APOA-5 Genetic Variant rs662799 with Metabolic Changes after an Intervention for 9 Months with a Low-Calorie Diet with a Mediterranean Profile"

_nutrients, 2022, doi:10.3390/nu14122427_

Round 1
Reviewer 1 Report
I have reviewed the manuscript titled ‘Association of APOA-5 Genetic Variant rs662799 with Meta-2 bolic Changes after the Intervention for 9 Months with a Low 3 Calorie Diet with a Mediterranean Profile”. Daniel-Antonio de Luis Roman., et al., studied the APOA-5 Genetic Variant rs662799 role in metabolic changes after low-calorie diet with Mediterranean pattern for 9 months in obese patients.
The study is designed with a bit lack of proper control groups i.e., healthy control patients were not included. The results are not presented either in table or graph, it makes very difficult to understand the results.
I have cross-checked the references provided by authors, found that the same authors have published an article in 2021, titled “APOA-5 Genetic Variant rs662799: Role in Lipid Changes and Insulin Resistance after a Mediterranean Diet in Caucasian Obese Subjects” https://doi.org/10.1155/2021/1257145. The present study is a slight incremental to the published one, by making only changes in diet composition. It makes this study is not novel to publish in nutrients journal.
Author Response
First, we would like to thank the comments of the Editor and reviewers We appreciate all your comments. We have revised the English and change some paragraphs. However, since English is not the authors' mother tongue, any improvement in the text made by the editorial team will be accepted.
All changes have been highlighted in red
Reviewer 1:
- The lack of a proper group has been included in limitation section (Fifthly, the absence of a healthy control group decreases the power of the results obtained with our design)
- Tables 3 and 4 have been included
- The article in 2021, titled “APOA-5 Genetic Variant rs662799: Role in Lipid Changes and Insulin Resistance after a Mediterranean Diet in Caucasian Obese Subjects” https://doi.org/10.1155/2021/1257145. This previous article is a different design with a different population (only 3 months of dietary intervention).A population of 363 Caucasian obese patients was enrolled. Participants were assigned to consume a calorie-restricted (daily 500-700 calories intake reduction) diet for 3 months. The balanced distribution of macronutrients in this diet was 50% of the calorie value from carbohydrates, 30% from lipids, and 20% from proteins. Percentage of fats was 55% from monounsaturated fats, 30% from saturated fats, and 15% from polyunsaturated fats.). In the new design, we demonstrate effects with a higher caloric restriction (1000 cal) and longer dietary intervention (9 months) with the present design.
Reviewer 2 Report
Dear Authors,
To enhance the readability and wider context for the audience, I have some points below which I believe should be addressed to make the paper stronger.
Abstract:
I suggest adding one sentence summarizing the results of the study - that is, the conclusions.
Introduction:
Information on the characteristics of the Mediterranean diet is missing in this section.
Subjects and Methods:
Information on the method of selecting the test group should be completed: How was the number of participants calculated? Did all the invited patients agree to participate? Have they all completed the study?A diet of 1000 kcal / day is almost impossible to balance with the supply of certain vitamins and minerals. Did the participants receive any additional vitamin and mineral supplementation? The use of the same energy level (1000 kcal / d) causes a different caloric deficit in women and men. Could this have influenced the results?
Results:
Table 1: How was the energy consumption assessed at the start of the study? Energy consumption at the beginning (0 time) was estimated at 1728(1823) kcal / d, which in the case of obese people is rather impossible - unless they were already on a reduction diet at the beginning. Can the authors explain this?
I am also wondering about the obtained results: with the consumption of 1000 kcal in obese people (BMI about 40) within 3 months (3 months vs 9 months) a weight loss of 2 kg was obtained, i.e. less than 1 kg per month. This is very little with such a large deficit. Can the authors comment on this?
The results are discussed for the whole group. Are there any possible gender related relationships in this case?
Conclusions:
Could the authors suggest any practical guidelines for implementation in this section?
Author Response
First, we would like to thank the comments of the Editor and reviewers We appreciate all your comments. We have revised the English and change some paragraphs. However, since English is not the authors' mother tongue, any improvement in the text made by the editorial team will be accepted.
All changes have been highlighted in red
Reviewer2:
Abstract:
We have included one sentence summarizing the results of the study “ In conclusion, non-C- allele carriers of rs662799 of APOA5 gene showed an improvement in triglyceride, insulin and HOMA-IR levels after low calorie diet with a Mediterranean pattern, with a lack of this effect in C allele carriers despite a significant weight loss. ”
Introduction:
Information on the characteristics of the Mediterranean diet has been included (The Mediterranean diet is a primarily plant-based eating plan that includes daily intake of whole grains, olive oil, fruits, vegetables and legumes, nuts, herbs, and spices. Other foods like animal proteins are eaten in smaller quantities, with the preferred animal protein being fish and seafood. These types of foods produce high amounts of dietary unsaturated fats and potential effects on lipid profile due to the presence of foods with a direct effect on lipids levels [15,16].)
Subjects and Methods:
Information on the method of selecting the test group should be completed. The selection method has been included “The present study enrolled a total of 269 unrelated obese subjects from our Health Area during routine check-up of obesity with a body mass index >30 kg/m2. The selection method was consecutive non-probabilistic.”
How was the number of participants calculated? We have included a sentence to explain this fact (Sample size (n=250) was obtained to determine differences over 10 mg/dl of triglyceride levels with 90% power and 5% significance.)
Did all the invited patients agree to participate? Yes, all patients agree.
Have they all completed the study? All subjects completed the study because to improve participant’s compliance during the whole study period, the dietitian interviewed them weekly by phone (see table 1).
Did the participants receive any additional vitamin and mineral supplementation? The participants did not receive vitamin and mineral, we thought that the intervention time was short.
The use of the same energy level (1000 kcal / d) causes a different caloric deficit in women and men. Could this have influenced the results? We have included this one as a new limitation “Sixthly, we used the same caloric restriction in males and females and perhaps this one is a potential bias, too”
Results:
Table 1: How was the energy consumption assessed at the start of the study? Energy consumption at the beginning (0 time) was estimated at 1728(1823) kcal / d, which in the case of obese people is rather impossible - unless they were already on a reduction diet at the beginning. Can the authors explain this?
The energy consumption was assessed as indicated in MyM section (All the subjects were given written instructions by a registered dietitian on completion of a 3-day (1-weekend and 2-week days) dietary record and basal time and after 3 and 9 months of dietary intervention. These records were analysed with a computer-based data evaluation system (Dietsource ®, Ge, Swi) with food tables reference of our country [25].) This is a potential limitation of our study and we included this fact in discussion (Firstly, dietary intake was based on self-reports obtained from patients, this self-reported data could overestimate or underestimate the actual dietary intake.)
I am also wondering about the obtained results: with the consumption of 1000 kcal in obese people (BMI about 40) within 3 months (3 months vs 9 months) a weight loss of 2 kg was obtained, i.e. less than 1 kg per month. This is very little with such a large deficit. Can the authors comment on this? The reviewer is absolutely right. A possible explanation for the slowing of weight loss between 6 and 9 months is a compensatory decrease in basal energy expenditure in obese patients as a saving mechanism.
The results are discussed for the whole group. Are there any possible gender related relationships in this case? We realized the study with both genders and the results were similar. And we used only the whole group, because TC+CC group had 32 patients, only 9 males. With this small group of males in the second group is very difficult to do a powerful statistical analysis.
Conclusions:
Could the authors suggest any practical guidelines for implementation in this section? We have included a new paragraph and a new reference “And in the future, clinical guidelines will incorporate all this knowledge to improve the design of nutritional interventions in obese patients, through panels of SNVs and other omic technologies [40].”
Reviewer 3 Report
Thanks for the opportunity to review this paper which aims to improve our understanding of the role of genetics in the body's response to various diet types and food components. Please find attached some comments for your consideration.

Author Response
First, we would like to thank the comments of the Editor and reviewers We appreciate all your comments. We have revised the English and change some paragraphs. However, since English is not the authors' mother tongue, any improvement in the text made by the editorial team will be accepted.
All changes have been highlighted in red
General comments · It is important to state that there were no controls in this study. We agree with the reviewer and we have included a new sentence in limitation paragraph “Fifthly, the absence of a healthy control group decreases the power of the results obtained with our design”
The association described is more of descriptive rather than causal. We agree with the comment of reviewer
We have revised references.
Table 3 and Table 4 have been included
- All these comments have been included)
Line Comment 26 Full stop after the word “loss”
64 obtain >> obtained 68,69 250 during 9 months >> during a period of 9 months during 3 months >> during a period of 3 months
75 old
188, 203 table >> Table
259 Other >> another
264, 265 In an intervention trial of 3 months with a hypocaloric Mediterranean diet in a Caucasian population, it has been reported [17] a decrease in… Consider rephrasing the statement to read: An intervention trial of 3 months with a hypocaloric Mediterranean diet in a Caucasian population [17] reported a decrease in…
271 higher than previous >> higher than the previous
272 more >> a higher
279 tatty >> fatty FAA >>FFA
287 response of >> response to
302 also influence >> also have an influence
314 Ethnic group >> sub-population In conclusion, this long-term study was able to demonstrate that the minor C allele of APOA5 gene (rs662799) is negatively related with the response of triglyceride, insulin levels and HOMA-IR after a low-calorie diet with Mediterranean pattern. Consider rephrasing the statement to read: In conclusion, this long-term study demonstrated that the minor C allele of the APOA5 gene (rs662799) is negatively related to the response with triglyceride, insulin levels, and HOMA-IR after a low-calorie diet with a Mediterranean patter
Round 2
Reviewer 1 Report
The manuscript by Daniel et al., titled. "Association of APOA-5 Genetic Variant rs662799 with Metabolic Changes after the Intervention for 9 Months with a Low Calorie Diet with a Mediterranean Profile".
This manuscript must be improved in representing results.
Reviewer 2 Report
Thanks to Authors for reviewing the manuscript